


# Atmospheric N$_2$O and CH$_4$ total columns retrieved from low-resolution FTIR spectra (Bruker Vertex 70) in the mid-infrared region

Minqiang Zhou[1,2], Bavo Langerock[2], Mahesh Kumar Sha[2], Christian Hermans[2], Nicolas Kumps[2], Rigel Kivi[3], Pauli Heikkinen[3], Christof Petri[4], Justus Notholt[4], Huilin Chen[5,6], and Martine De Mazière[2]

[1]CNRC & LAGEO, Institute of Atmospheric Physics, Chinese Academy of Sciences, Beijing, China
[2]Royal Belgian Institute for Space Aeronomy (BIRA-IASB), Brussels, Belgium
[3]Finnish Meteorological Institute, Space and Earth Observation Centre, Sodankylä, Finland
[4]Institute of Environmental Physics, University of Bremen, Bremen, Germany
[5]Centre for Isotope Research (CIO), Energy and Sustainability Research Institute Groningen (ESRIG), University of Groningen, Groningen, the Netherlands
[6]Joint International Research Laboratory of Atmospheric and Earth System Sciences, School of Atmospheric Sciences, Nanjing University, Nanjing, China

**Correspondence:** Minqiang Zhou (minqiang.zhou@mail.iap.ac.cn; minqiang.zhou@aeronomie.be)

**Abstract.** Nitrous oxide (N$_2$O) and Methane (CH$_4$) are two important greenhouse gases in the atmosphere. In 2019, mid-infrared (MIR) solar absorption spectra were recorded by a Bruker Vertex 70 spectrometer and a Bruker IFS 125HR spectrometer at Sodankylä, Finland at spectral resolutions of 0.2 cm$^{-1}$ and 0.005 cm$^{-1}$, respectively. The N$_2$O and the CH$_4$ retrievals from high-resolution MIR spectra have been well investigated within the Network for Detection of Atmospheric Composition Change (NDACC), but not for MIR spectra gathered with instruments operating at low spectral resolution. In this study, N$_2$O and CH$_4$ retrieval strategies and retrieval uncertainties from the Vertex 70 MIR low-resolution spectra are discussed and presented. The accuracy and precision of the Vertex 70 N$_2$O and CH$_4$ retrievals are assessed by comparing them with the co-located 125HR retrievals and AirCore measurements. The relative differences between the N$_2$O total columns retrieved from 125HR and Vertex 70 spectra are -0.3±0.7(1$\sigma$)% with a correlation coefficient (R) of 0.93. Regarding CH$_4$ total column, we first used the same retrieval microwindows for 125HR and Vertex 70 spectra, but there is an underestimation in the Vertex 70 retrievals, especially in summer. The relative differences between the CH$_4$ total columns retrieved from the 125HR and Vertex spectra are -1.3±1.1% with a R value of 0.77. To improve the Vertex 70 CH$_4$ retrievals, we propose an alternative retrieval microwindows. The relative differences between the CH$_4$ total columns retrieved from the 125HR and Vertex spectra in these new windows become 0.0±0.8%, along with an increase in R value to 0.87. The co-located AirCore measurements confirm that the Vertex 70 CH4 retrievals using the latter window choice are better, with the relative mean differences between the Vertex CH4$_4$ retrievals and AirCore measurements of -1.9% for the standard NDACC mircrowindows, and of 0.13% for the alternative microwindows. This study provides an insight into the N$_2$O and CH$_4$ retrievals from the low-resolution (0.2 cm$^{-1}$) MIR spectra observed with a Vertex 70 spectrometer, and demonstrates the suitability of this kind of instruments for contributing to satellite validation, model verification, and other scientific campaigns with the advantage of their transportability and lower cost compared to standard NDACC-type FTIR instruments.



## 1 Introduction

Nitrous oxide ($N_2O$) and methane ($CH_4$) are two important atmospheric greenhouse gases, which contribute greatly to global climate change (IPCC, 2013). The global mean $N_2O$ concentration is increasing continuously since the pre-industrial era and reaches up to 332.87 ppb in 2020 as observed by the National Oceanic and Atmospheric Administration (NOAA; https://gml.noaa.gov/ccgg/trends_n2o/) surface measurements (Hall et al., 2007). Moreover, the NOAA surface measurements show that the global annual growth rate of $N_2O$ is increasing recently. The $N_2O$ annual growth rate is about 1.0 ppb/year during the last five years (2015-2020) compared to that of 0.9 ppb/year from 2010 to 2015. The increasing $N_2O$ concentration also contributes to stratospheric ozone depletion (Park et al., 2012). Tian et al. (2020) used both bottom-up and top-down methods to estimate the atmospheric $N_2O$ emissions between 1980 and 2016, and highlighted that it is urgent to mitigate the human-induced $N_2O$ emissions, as its measured growth rate exceeds some of the highest projected emission scenarios. According to the surface in situ measurements (WMO, 2020), the global mean $CH_4$ concentration kept increasing from 1984 to 1999, and remained stable between 1999 and 2006, but started to increase again after 2007. It is not fully understood what caused the variation of $CH_4$ during the last decades, as the sources of atmospheric $CH_4$ are diverse and variable (Saunois et al., 2020). Many factors can affect the atmospheric $CH_4$ concentration, such as the natural wetland and fossil fuel emissions (Kirschke et al., 2013), the tropical biogenic sources (Schwietzke et al., 2016), the biomass burning emissions (Worden et al., 2017), and the atmospheric OH level (Turner et al., 2017).

Mid-infrared (MIR) solar absorption spectra (2400-3000 $cm^{-1}$) recorded by the ground-based Fourier-transform infrared (FTIR) spectrometer are used to retrieve the $N_2O$ and $CH_4$ vertical profiles and total columns within the Network for Detection of Atmospheric Composition Change - InfraRed Working Group (NDACC-IRWG) (De Mazière et al., 2018). The NDACC $N_2O$ and $CH_4$ measurements are widely used for understanding their temporal variations (Angelbratt et al., 2011; Zhou et al., 2018), for satellite validation (Barret et al., 2021; Sha et al., 2021), and for model comparison and verification (Bader et al., 2017; Zhou et al., 2019b). The NDACC-IRWG sites are typically operated with either a Bomem DA8, a Bruker IFS 120 M/HR, or a Bruker IFS 125 M/HR instrument to record the MIR spectra with a high spectral resolution of 0.003-0.005 $cm^{-1}$. The lower spectral resolution spectrometers, such as a Bruker Vertex 70, a Bruker EM27/SUN, and a Bruker IRcube with a spectral resolution of 0.16-0.5 $cm^{-1}$, have advantages such as lower-cost, higher temporal resolution, and transportability.

Within the Fiducial Reference Measurements for Ground-Based Infrared Greenhouse Gas Observations (FRM4GHG) project, carbon dioxide ($CO_2$), methane ($CH_4$), and carbon monoxide (CO) total columns retrieved from the near-infrared (NIR) solar spectra recorded by these lower spectral resolution spectrometers are proved to be comparable with the data products retrieved from the Total Carbon Column Observing Network (TCCON) high spectral-resolution measurements (Sha et al., 2020). Inter-comparisons between low and high resolution MIR spectra and retrievals have been investigated in the past as well, e.g., Taylor et al. (2008) used two FTIRs with differing spectral resolution (0.004 $cm^{-1}$ for the Bomen DA8 and 0.2 $cm^{-1}$ for the ABB Analytical DA5) over a period of four months in the summer of 2005, but they pointed out that the relatively poor correlations were observed for the $N_2O$ and $CH_4$ column observed with high and low spectral resolutions, with the median differences are -0.36% for $N_2O$ and 3.7% for $CH_4$.





The Bruker Vertex 70 instrument is capable of measuring NIR and MIR spectra quasi-simultaneously. However, to our knowledge, the retrievals of $N_2O$ and $CH_4$ from the MIR spectra observed by the Bruker Vertex 70 instrument have not yet been investigated. To better understand the $N_2O$ and $CH_4$ retrievals from the MIR spectra observed by lower spectral resolution measurements, we operated the Bruker Vertex 70 and the Bruker 125HR next to each other at Sodankylä in 2019 (March

- November). The spectral resolutions of the Bruker Vertex 70 and the Bruker IFS 125HR are 0.2 cm$^{-1}$ and 0.005 cm$^{-1}$, respectively. Further details of the instrumental set-up are provided in Section 2. The retrieval is performed with the SFIT4 v0.9.4.4 code (Pougatchev et al., 1995), which is commonly used in the NDACC-IRWG community. The retrieval strategies and retrieval uncertainties of $N_2O$ and $CH_4$ are discussed in Section 3. In Section 4, the $N_2O$ and $CH_4$ retrievals derived from the Vertex 70 are compared to those from the 125HR measurements following NDACC recommended retrieval strategies. In

addition, the $CH_4$ retrievals are also compared to the co-located AirCore profiles. Finally, the conclusions are drawn in Section 5.

## 2   Site description and instrument configuration

The FTIR measurements used in this study are carried out at the Sodankylä facility of the Finnish Meteorological Institute (FMI), North Finland (67.4°N, 26.6°E, 180 m a.s.l.). The FMI site is located in a boreal forest region, and is about 6 km

south of the downtown. Due to the high latitude, the site is located in the polar night in boreal winter, and the atmospheric components above the site are affected by polar vortex conditions in winter and spring (Kivi et al., 2007; Ostler et al., 2014).

A Bruker IFS 125HR FTIR instrument has been operated at Sodankylä since 2009. The 125HR initially only recorded NIR spectra (4000 - 15000 cm$^{-1}$) with Indium Gallium Arsenide (InGaAs) and Silicon (Si) detectors, and has been affiliated to the TCCON network (Kivi and Heikkinen, 2016; Wunch et al., 2015). In 2011, the 125HR started measuring MIR spectra (2200-

3500 cm$^{-1}$) using an Indium Antimonide (InSb) detector as well, with a spectral resolution of 0.005 cm$^{-1}$. The NIR and MIR spectra at Sodankylä have been observed quasi-simultaneously since then. Profiles of $N_2O$ and $CH_4$ have been successfully retrieved from the MIR spectra using the SFIT4 code following the NDACC-IRWG guide, and the products are supporting the Copernicus Atmosphere Monitoring Service (https://cams27.aeronomie.be/), and the TROPOMI satellite validation (Sha et al., 2021).

During the ESA-funded FRM4GHG project, several portable FTIR instruments (Bruker EM27/SUN, Bruker IRcube and Bruker Vertex 70) were operated between 2017 and 2019 close to the Bruker 125HR instrument at Sodankylä. The schematic diagram of the Vertex 70 measurement is shown in Figure 1. In 2017 and 2018, only NIR spectra were recorded by the Vertex70, because the main objective of this campaign was to study the performance of $CO_2$, $CH_4$ and CO retrieved from the NIR spectra observed by the portable FTIRs, and compared them to the TCCON NIR measurements (Sha et al., 2020). However, it is also

important to understand the performance of $N_2O$ and $CH_4$ retrievals derived from the MIR spectra observed by the portable FTIR and to compare them with the NDACC MIR measurements performed at a high spectral resolution. Therefore, at the beginning of 2019, we added an InSb detector in the Vertex 70 instrument, and performed one-year of measurements to record





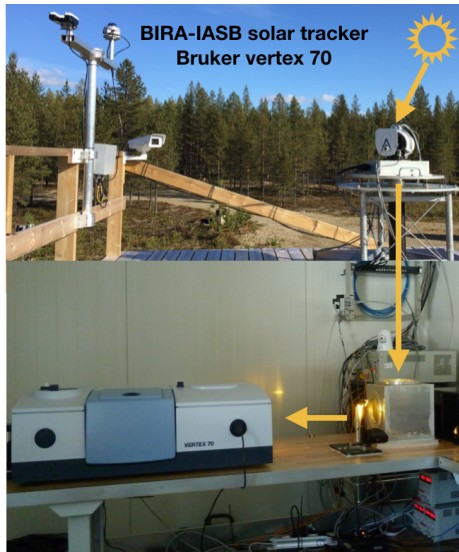

**Figure 1.** The Bruker Vertex 70 system operated at Sodankylä during the FRM4GHG campaign, including the weather station, the solar tracker. The orange arrows shows the light path coming from the sun till it enters the FTIR instrument.

MIR spectra. Table 1 lists the main characteristics of the MIR spectra observed by the Vertex 70 and 125HR instruments, more information about the instruments and the FRM4GHG Sodankylä campaign can be found in Sha et al. (2021).

**Table 1.** The main characteristics of the spectra observed by Vertex 70 and 125HR at Sodankylä used in this study.

|  | Vertex 70 | 125HR |
| --- | --- | --- |
| Detector | Insb | |
| Spectral range | $2200\text{-}3500\ \text{cm}^{-1}$ | |
| Spectral resolution | $0.2\ \text{cm}^{-1}$ | $0.005\ \text{cm}^{-1}$ |
| Number of scan | 12 | 2 |
| Duration of one measurement | 52 s | 194 s |

## 3  FTIR retrievals

$N_2O$ and $CH_4$ partial column profiles are derived from the width and shape of their absorption lines in the solar spectra ($\boldsymbol{y}$) observed by the ground-based FTIR instrument, modeled as

$$\boldsymbol{y} = \mathbf{F}(\boldsymbol{x}, \boldsymbol{b}) + \boldsymbol{\varepsilon}, \tag{1}$$

where $\mathbf{F}$ is the forward atmospheric radiation transfer model, $\boldsymbol{x}$ is the state vector (the retrieved parameters), $\boldsymbol{b}$ is the forward model parameters but not retrieved, and $\boldsymbol{\varepsilon}$ represents the measurement noise and forward model uncertainty. In this study, the





SFIT4 algorithm is used to simulate the observed spectrum and to minimize the residual between the simulated and the observed spectra by looking for the optimal state vector using an inverse model (optimal estimation model or Tikhonov regularization). The inverse problem is solved by using the Levenberg-Marquardt Gauss-Newton iteration in the SFIT4 algorithm, and the retrieved state vector can be related to the actual state of the atmosphere by

$$\boldsymbol{x}_r = \boldsymbol{x}_a + \mathbf{A}(\boldsymbol{x}_t - \boldsymbol{x}_a) + \boldsymbol{\varepsilon}, \tag{2}$$

where $\boldsymbol{x}_a$ and $\boldsymbol{x}_t$ are the a priori and the true state vectors, respectively. $\mathbf{A}$ is the averaging kernel matrix (AVK), representing the sensitivity of the retrieved vertical profile of the target species to the true profile. $\boldsymbol{\varepsilon}$ is the retrieval uncertainty. The trace of the $\mathbf{A}$ is the degrees of freedom (DOF), indicating the number of independent pieces of information (Rodgers, 2000). Following the Eq. 2, the retrieved total column can be written as

$$\boldsymbol{TC}_r = \boldsymbol{TC}_a + \boldsymbol{A_{col}} \cdot (\boldsymbol{PC}_t - \boldsymbol{PC}_a) + \boldsymbol{\varepsilon}, \tag{3}$$

where $\boldsymbol{TC}_a$ and $\boldsymbol{TC}_r$ are the a priori and retrieved total columns, respectively. $\boldsymbol{PC}_a$ and $\boldsymbol{PC}_t$ are the a priori and true partial columns, respectively. $\boldsymbol{A_{col}}$ is the column averaging kernel (CAVK), representing the sensitivity of the retrieved total column to the true partial column.

## 3.1 Retrieval strategy

### 3.1.1 N$_2$O

The N$_2$O retrieval strategy for the high spectral resolution measurements from 125HR is well harmonized within the NDACC-IRWG community (Zhou et al., 2019b). Four micro-windows (2481.3-2482.6, 2526.4-2528.2, 2537.85-2538.8, 2540.1-2540.7 cm$^{-1}$) including the N$_2$O absorption lines are used to retrieve the N$_2$O vertical profile (Figure 2), and the HITRAN2008 spectroscopy is adopted (Rothman et al., 2009). To reduce the effect from the interfering species, HDO, CO$_2$ and CH$_4$ columns are retrieved simultaneously together with the N$_2$O vertical profile. The a priori profiles of the target species and CO$_2$ and CH$_4$

interfering species are created from the Whole Atmosphere Community Climate Model (WACCM) v4 monthly means between 1980 and 2020. Due to its high spatial-temporal variability, the temperature vertical profile and the a priori profile of HDO is derived from the National Centers for Environmental Prediction (NCEP) 6-hourly reanalysis data with interpolation to the measurement time. In this study, the same four windows are used for Vertex 70 and 125HR N$_2$O retrievals. The signal-to-noise

ratio (SNR) of the spectra is calculated from the observed spectra, and then the measurement uncertainty $\mathbf{S}_\epsilon$ is calculated as $1/\text{SNR}^2$ for the diagonal values and 0 for the off-diagonal values. The regularization matrix $\mathbf{R}$ for the N$_2$O retrieval is created with the Tikhonov $\mathbf{L}_1$ method $\mathbf{R} = \alpha \mathbf{L_1}^T \mathbf{L_1}$ (Tikhonov, 1963), and the same regulation matrix is applied to Vertex 70 and 125HR retrievals. To determine the regularisation strength $\alpha$, we tune it so that the retrieved DOF is similar to the DOF obtained when retrieving N$_2$O with the optimal estimation method (Steck, 2002) and where the a priori covariance matrix

30

(**Sa**) is derived from the WACCM model monthly means. The DOF of N$_2$O profile retrievals from 125HR spectra is about 2.5 derived by the OEM, so that we set the $\alpha$ to 4000 to get a similar DOF with the Tikhonov method.



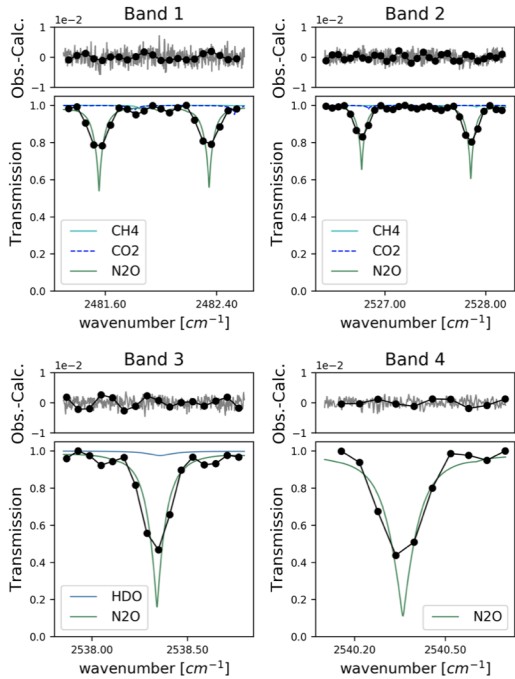

**Figure 2.** The four mirco-windows used for $N_2O$ retrieval. In each band, the transmittances from atmospheric species, together with the Vertex 70 typical spectra are shown in the lower panel, and the typical residuals for the Vertex 70 (black) and 125HR (grey) spectra at Sodankylä are presented in the upper panel.

### 3.1.2 $CH_4$

First, we apply the standard NDACC retrieval strategy for $CH_4$ from 125HR spectra. Six microwindows (2611.6-2613.35, 2613.7-2615.4, 2835.55-2835.8, 2903.82-2903.925, 2914.7-2915.15 and 2941.51-2942.22 $cm^{-1}$; Bands 1-6 ) are used (Sepúlveda et al., 2014; Zhou et al., 2018), and the columns of $H_2O$, HDO, $CO_2$ are retrieved together with the $CH_4$ profile. Figure 3 shows the typical transmittances from the atmospheric species for $CH_4$ retrievals at Sodankylä, together with the fitting residuals for Vertex 70 and 125HR spectra in each window. However, there are only 4 and 2 wavenumbers in bands 3 and 4, respectively, for the Vertex 70 low spectral resolution spectra, and the signals in bands 3 and 4 for the Vertex low spectral resolution spectra are 2-3 times weaker as compared to the high spectral resolution spectra. Therefore, we also apply two other relative broad windows, including stronger $CH_4$ absorption lines to provide more information. As a result, we apply two window settings for Vertex 70 $CH_4$ retrieval: i) band 1 - band 6 (same as the 125HR), named V1; ii) replacing band 3 and band 4 with band 3V2 (2875.15-2875.85 $cm^{-1}$) and band 4V2 (2898.3-2898.93 $cm^{-1}$), named V2. Regarding the spectroscopic data, several line lists have been tested for $CH_4$ retrievals within the NDACC-IWRG community. Currently, the ATM19 and HITRAN01 are two recommended line lists at present (Chesnokova et al., 2020). In this study, we take the atmospheric line list ATM19 for $CH_4$ retrieval. The fitting residuals in 3V2 and 4V2 bands are comparable to other windows (bands 1,2,5,6) for Vertex 70.





In fact, we have tested many mircowindows in addition to bands 3V2 and 4V2, but these two bands offer us the best result (close to 125HR retrievals; see Section 4.2). Note that the observed spectrum with a high spectral resolution (125HR) is not well fitted in band 3V2. Therefore, we do not apply the V2 window choice for 125HR spectra. As for the $N_2O$ retrieval, the a priori profiles of $CH_4$,$CO_2$ are created as the average WACCM monthly mean between 1980 and 2020, and the a priori profile of $H_2O$ and HDO are derived from the NCEP 6-hourly reanalysis data with interpolation to the measurement time. The SNR of the spectra is calculated from the observations. The regularization matrix $\mathbf{R}$ for the $CH_4$ retrieval is also created with the Tikhonov $\mathbf{L}_1$ method, with the $\alpha = 10000$, leading to a mean DOF of 2.1 for 125HR retrievals.

## 3.2 Retrieval vertical sensitivity

### 3.2.1 $N_2O$

Figure 4 shows both the CAVK and AVK together with the a priori and retrieved $N_2O$ vertical profiles for 125HR and Vertex 70. The mean and standard deviation (std) of the DOF are 2.6±0.1 for 125HR and 1.6±0.1 for Vertex 70. As expected, more vertical information is derived from the high spectral resolution spectra (125HR) as compared to the low spectral resolution spectra (Vertex 70). The typical AVKs show that the retrieved $N_2O$ profile from 125HR has a good sensitivity between the surface and 25 km, while the retrieved $N_2O$ profile from Vertex 70 is mainly sensitive to the vertical range between the surface and 20 km. The CAVK of 125HR is close to 1 in the whole vertical range and hardly varies with solar zenith angle (SZA). The CAVK of Vertex 70 is close to 1 in the whole vertical range at a large SZA (> 70°), but it starts deviating from 1 above 20 km at a relatively small SZA. Since the partial column of $N_2O$ above 20 km takes up only about 1.5% of the total column, such deviation in CAVK of Vertex does not affect the retrieved $N_2O$ total column strongly, and the smoothing error of the $N_2O$ total column is very small (see Section 3.3). The retrieved $N_2O$ vertical profiles from 125HR and Vertex 70 are similar, but not exactly the same. Due to the limited vertical information, we will focus on the partial columns or the total column instead of the vertical profile (see Section 4).

### 3.2.2 $CH_4$

The CAVK, AVK and retrieved profiles from 125HR and Vertex 70 (V1 and V2) are shown in Figure 5. The vertical sensitivities of the retrieved $CH_4$ total column from 125HR and Vertex 70 V1 are similar, since they use the same retrieval windows. There is no obvious SZA dependence in CAVK for the 125HR $CH_4$ retrievals. However, for both Vertex V1 and V2 retrievals, the CAVK varies with SZA, especially above 10 km. The mean DOF from 125HR $CH_4$ retrievals is 2.1, which is about twice than that from the Vertex V1 retrievals (DOF ~1.0). As for the $N_2O$ retrievals, more vertical information can be obtained with a higher spectral resolution. The CAVK of the Vertex 70 V2 retrieval shows that the retrieved $CH_4$ total column has better sensitivity in the lower troposphere as compared to the Vertex V1 retrieval, and the mean DOF is 1.3. The vertical information obtained in the Vertex V2 retrieval is higher than that in the Vertex V1 retrieval. Still, the DOFs indicate that we must focus on the $CH_4$ total columns for both the Vertex V1 and Vertex V2 retrievals.

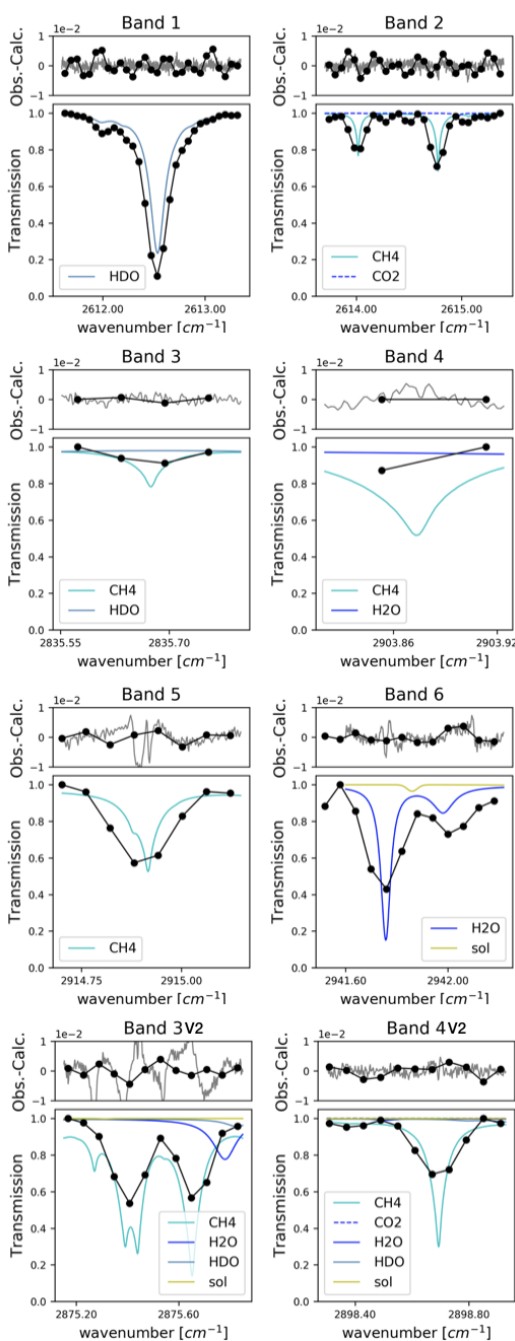

**Figure 3.** The six mirco-windows (Bands 1-6) used for 125HR and Vertex 70 V1 $CH_4$ retrieval, together with the other 2 mirco-windows used for Vertex 70 V2 $CH_4$ retrieval. In each band, the transmittances from atmospheric species, together with the Vertex 70 typical spectra are shown in the lower panel, and the typical residuals for the Vertex 70 (black) and 125HR (grey) spectra at Sodankylä are presented in the upper panel.



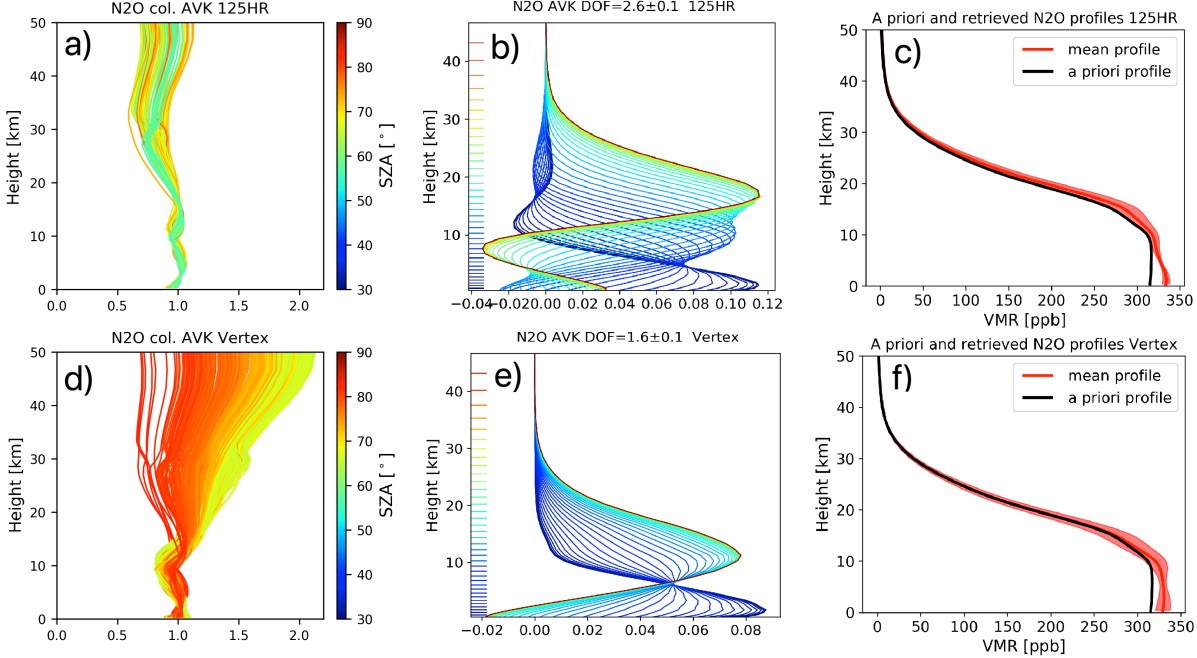

**Figure 4.** $N_2O$ retrievals: the column averaging kernels' variation with solar zenith angles (in unit of $(mol/cm^2)/(mol/cm^2)$), the typical averaging kernel matrix (in unit of $(vmr/mvr)/(vmr/vmr)$), and the a priori profile (black) and retrieved profiles (mean: red solid line; $1\sigma$: red shaded area) between the surface and 50 km for 125HR retrievals (a, b, c) and for Vertex 70 retrievals (d, e, f).

### 3.3 Retrieval uncertainty

According to Rodgers (2000), the retrieval uncertainty of $\boldsymbol{x}_r$ can be written as

$$\boldsymbol{x}_r - \boldsymbol{x}_t = (\mathbf{A} - \mathbf{I})(\boldsymbol{x}_t - \boldsymbol{x}_a) + \mathbf{G}_y \mathbf{K}_b(\boldsymbol{b}_t - \hat{\boldsymbol{b}}) + \mathbf{G}_y \epsilon_{noise}, \tag{4}$$

where $\mathbf{G}_y$ is the contribution matrix, $\mathbf{K}$ is the Jacobian matrix, $\hat{\boldsymbol{b}}$ is the best estimate of the forward model parameter, $\boldsymbol{b}_t$ is
5   the true forward model parameter, and $\epsilon_{noise}$ is the noise of the observed spectra. The items on the right side of Eq. 4 are the smoothing error (first), the model parameter error (second) and the measurement error (third), respectively.

Regarding the smoothing error, the systematic uncertainty of $(\boldsymbol{x}_t - \boldsymbol{x}_a)$ for both $N_2O$ and $CH_4$ is set to 5%, and the random uncertainty of $(\boldsymbol{x}_t - \boldsymbol{x}_a)$ is created from the covariance matrix from the WACCM model monthly means between 1980 and 2020. The important model parameters are the spectroscopy, temperature, SZA and zero level shift (zshift). The systematic
10   uncertainties of the $N_2O$ and $CH_4$ line intensities are set to 3%, and the systematic uncertainty of the spectroscopic temperature and pressure broadening parameters are set to 5% (Rothman et al., 2009; Gordon et al., 2017). It is assumed that there is no random uncertainty associated with the spectroscopic parameters. The systematic and random uncertainties of the temperature profiles are evaluated by calculating the mean differences and std, respectively, of ERA5 and NCEP temperature profiles in 2019. The systematic and random uncertainties of SZA are set to 0.1% and 0.5%, respectively. The systematic and random

**Figure 5.** CH$_4$ retrievals: the column averaging kernels' variation with solar zenith angles (in unit of $(mol/cm^2)/(mol/cm^2)$), the typical averaging kernel matrix (in unit of $(vmr/mvr)/(vmr/vmr)$), and the a priori profile (black) and retrieved profiles (mean: red solid line; $1\sigma$: red shaded area) between the surface and 50 km for 125HR retrievals (a, b, c), for Vertex V1 (d, e, f) and Vertex V2 (g, h, i) retrievals.



uncertainties of zshift are both set to 0.1%. We assume that the measurement noise has only a random uncertainty component and evaluate it as $\epsilon_{noise} = 1/\text{SNR}$, with SNR the calculated signal-to-noise ratio of the spectra.

The systematic and random retrieval uncertainties of the retrieved $N_2O$ and $CH_4$ profiles retrieved from 125HR and Vertex 70 spectra are then evaluated according to Eq.4 with the above assuptions and are listed in Tables 2 and 3, respectively. The

5 systematic and random retrieval uncertainties of the $N_2O$ total column are 3.6% and 1.5% for retrievals from 125HR spectra, and 3.7% and 1.8% for retrievals from Vertex 70 spectra. The systematic and random retrieval uncertainties of the $CH_4$ total column are 3.9% and 1.6% for 125HR spectra retrievals, 4.1% and 2.3% for Vertex 70 spectra V1 retrievals, and 4.3% and 2.4% for Vertex 70 spectra V2 retrievals. For both $N_2O$ and $CH_4$ retrievals, the dominant systematic uncertainties are coming from the spectroscopy. According to our estimations, the random uncertainties in $N_2O$ retrievals between 125HR and Vertex 70

are dominated by the measurement error, while the random uncertainties in $CH_4$ retrievals from 125HR and Vertex 70 spectra are dominated by the temperature, zshift and the measurement errors. To assess the estimated random uncertainty, we calculate the mean of all the stds of the retrieved $N_2O$ or $CH_4$ total columns within $\pm$ 1 hour around noon (at least 2 available retrievals) on each measurement day. We find a good agreement with the estimated random uncertainties for both 125HR and Vertex 70 retrievals, indicating that our estimated random uncertainties are reasonable. The estimated random uncertainties of retrievals

from 125HR spectra are smaller than those from Vertex 70 spectra for both $N_2O$ and $CH_4$ retrievals, and this is confirmed by the stds of the retrieved total columns.

**Table 2.** The relative uncertainties [%] of $N_2O$ total column retrievals from 125HR and Vertex 70 spectra at Sodankylä. '-' indicates zero uncertainty contribution. See tet for the interpretation of std [$\pm$1h noon].

| | 125HR | | Vertex 70 | |
| --- | --- | --- | --- | --- |
| | Systematic | Random | Systematic | Random |
| Smoothing | 0.1 | 0.1 | 0.1 | 0.2 |
| Spectroscopy | 3.5 | - | 3.6 | - |
| Temperature | 0.3 | 0.1 | 0.3 | 0.1 |
| SZA | 0.3 | 1.2 | 0.3 | 1.2 |
| zshift | 0.2 | 0.2 | 0.2 | 0.2 |
| Measurement | - | 0.1 | - | 0.9 |
| Total | 3.6 | 1.5 | 3.7 | 1.8 |
| std [$\pm$1h noon] | | 1.2 | | 2.0 |





**Table 3.** Same as Table 2, but for the relative uncertainties [%] of CH$_4$ total columns from 125HR and Vertex 70 V1 and V2 retrievals at Sodankylä.

|  | 125HR | | Vertex 70 V1 | | Vertex 70 V2 | |
| --- | --- | --- | --- | --- | --- | --- |
|  | Systematic | Random | Systematic | Random | Systematic | Random |
| Smoothing | 0.1 | 0.3 | 0.2 | 0.5 | 0.1 | 0.1 |
| Spectroscopy | 3.6 | - | 3.6 | - | 3.6 | - |
| Temperature | 1.0 | 0.3 | 1.7 | 1.6 | 1.5 | 1.5 |
| SZA | 0.3 | 1.2 | 0.2 | 1.0 | 0.2 | 1.0 |
| zshift | 0.9 | 0.9 | 1.0 | 1.0 | 1.4 | 1.4 |
| Measurement | - | 0.1 | - | 0.6 | - | 0.6 |
| Total | 3.9 | 1.6 | 4.1 | 2.3 | 4.3 | 2.4 |
| std [±1h noon] |  | 1.9 |  | 2.4 |  | 2.6 |

## 4  Results and discussions

### 4.1  N$_2$O

#### 4.1.1  Total column

The time series of all the individual N$_2$O total columns retrieved from 125HR and Vertex 70 are shown in Figure 6, together
with the relative differences ((Vertex-125HR)/125HR × 100%) between the co-located hourly means. The mean difference is -0.3% (1$\sigma$=0.7%), which is within the retrieval uncertainties of 125HR and Vertex (Table 2). To reduce the impact of the vertical sensitivities from 125HR and Vertex, we smooth the retrieved N$_2$O 125HR profile by the Vertex 70 averaging kernel. The results with and without smoothing correction are very close to each other, with a relative difference of less than 0.1%, because the smoothing error is very small. Low N$_2$O total columns in spring are observed both by 125HR and Vertex 70
measurements. There is no clear seasonal variation in the relative differences. Figure 7 shows that the Pearson correlation coefficient (R) between the co-located daily means is 0.93. The correlation coefficient R between the H$_2$O total columns and the relative differences is -0.02, and between the SZAs and the relative differences is 0.05, thus demonstrating that there are no water vapour or SZA dependences in the relative differences.

#### 4.1.2  Partial columns

Apart from the total column, we also compare the 125HR and Vertex 70 N$_2$O retrievals in two partial columns (lower layer: 0-6 km and upper layer: 6-25 km). The two partial columns are selected as we can get a DOF of 0.8 in each layer for the Vertex 70 N$_2$O retrievals. The DOFs of the 125HR N$_2$O are 1.0 and 1.5 for the lower and upper partial columns, respectively.





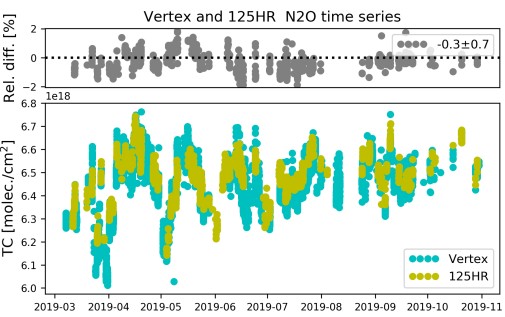

**Figure 6.** The time series of the N$_2$O total columns retrieved from 125HR and Vertex 70, together with the differences from their co-located hourly means.

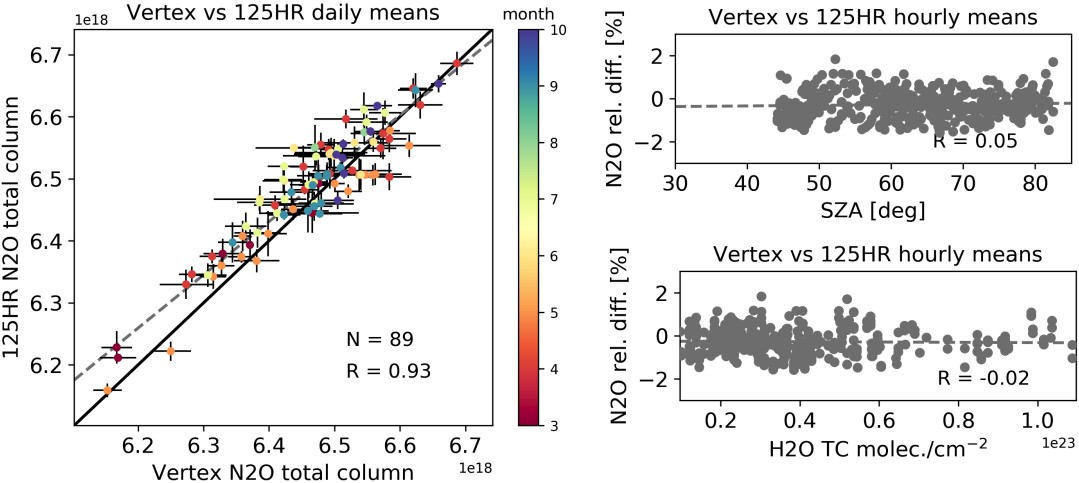

**Figure 7.** Left: the correlation plot between the co-located 125HR and Vertex 70 retrieved N$_2$O daily mean total columns, colored by the measurement month. The error bar represents the daily std. Right: the variations of the hourly relative differences with the water vapour total column and SZA. R is the correlation coefficient and N is the number of co-located days.





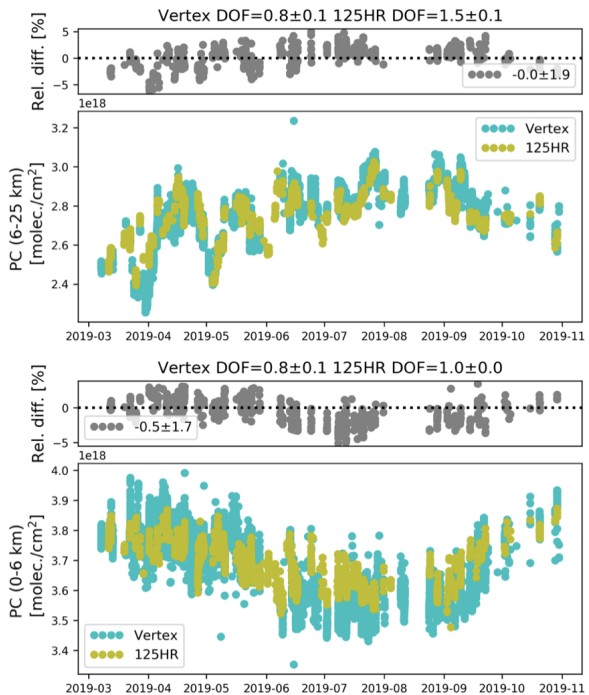

**Figure 8.** Same as Figure 6, but for $N_2O$ in two partial columns (lower: 0-6 km; uppder: 6-25 km).

The correction coefficients between the 125HR and Vertex 70 co-located $N_2O$ partial column daily means are 0.86 and 0.94 for the lower and upper layers, respectively. The time series of the $N_2O$ partial columns retrieved from 125HR and Vertex 70, together with their relative differences are shown in Figure 8. Seasonal variations of $N_2O$ partial columns observed by 125HR and Vertex 70 retrievals are consistent in both layers. The low $N_2O$ values in the upper layer due to the polar vertex in spring (Zhou et al., 2019b) are well captured by 125HR and Vertex 70 retrievals. The mean and std of the differences between the 125HR and Vertex is 0.0±1.9% in the partial column between 6 and 25 km, and -0.5±1.7% between the surface and 6 km. The mean differences in the partial columns are close to that in the total column. However, the stds of the differences in the partial columns are larger than that in the total column. What's more, it is found that there is a seasonal variation in the relative differences for both layers, e.g., the 125HR retrieved $N_2O$ is larger/lower than the Vertex retrieved $N_2O$ in spring/summer between 6 and 25 km. As the seasonal variations of the relative differences at the two layers are opposite, they compensate each other in the total column for which there is almost no seasonal variation of the relative difference. In summary, the partial columns (0-6 km and 6-25 km) of $N_2O$ can be retrieved with confidence from the Vertex 70 spectra, albeit with a larger uncertainty as compared to the 125HR retrievals.





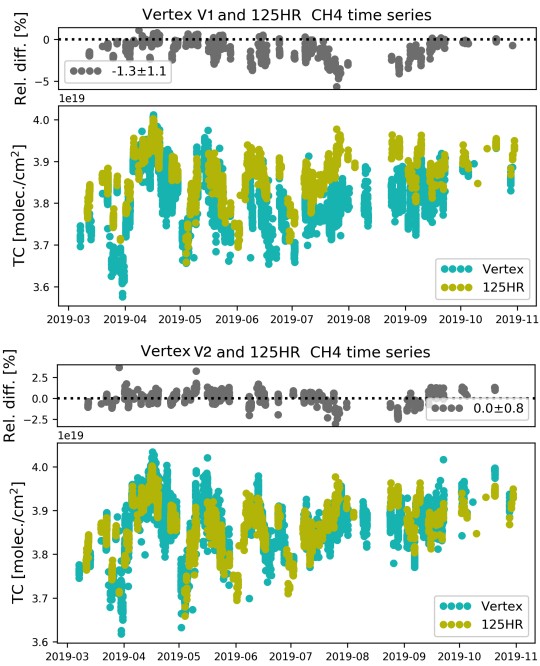

**Figure 9.** Same as Figure 6, but for the comparison between 125HR and Vertex 70 V1 $CH_4$ total columns (upper), and between 125HR and Vertex 70 V2 $CH_4$ total columns (lower).

## 4.2 $CH_4$

### 4.2.1 Total column

Since the mean DOF of the Vertex $CH_4$ retrievals is relatively small (1.0 for V1 and 1.3 for V2), we only look in this section only at the $CH_4$ total columns retrieved from 125HR and Vertex 70 spectra. As for the previous comparisons, the 125HR $CH_4$

5  profile is smoothed with the Vertex 70 AVK to reduce the impact of the different vertical sensitivities of 125HR and Vertex retrievals. The time series of the retrieved $CH_4$ total columns from 125HR, Vertex 70 V1 and V2 are shown in Figure 9. The mean difference between the 125HR and Vertex 70 V1 $CH_4$ total columns is -1.3±1.1%. The Vertex V1 $CH_4$ total column is much lower than the 125HR retrieved $CH_4$ in summer. The mean difference between the 125HR and Vertex V2 $CH_4$ total columns is 0.0±0.8%.

10  The scatter plots (Figure 10) confirm that the Vertex V2 $CH_4$ total columns are much closer to the 125HR retrievals as compared to the Vertex V1 results. The correlation coefficients between 125HR and Vertex 70 co-located $CH_4$ total column daily means are 0.77 for Vertex V1 and 0.87 for Vertex V2. The relative differences between 125HR and Vertex V1 $CH_4$ total columns are varying with SZA (R=0.42) and $H_2O$ total column (R=-0.76). The relative differences between 125HR and Vertex V2 $CH_4$ total columns have almost no SZA dependence (R=-0.03), and the correlation with the $H_2O$ total column is strongly

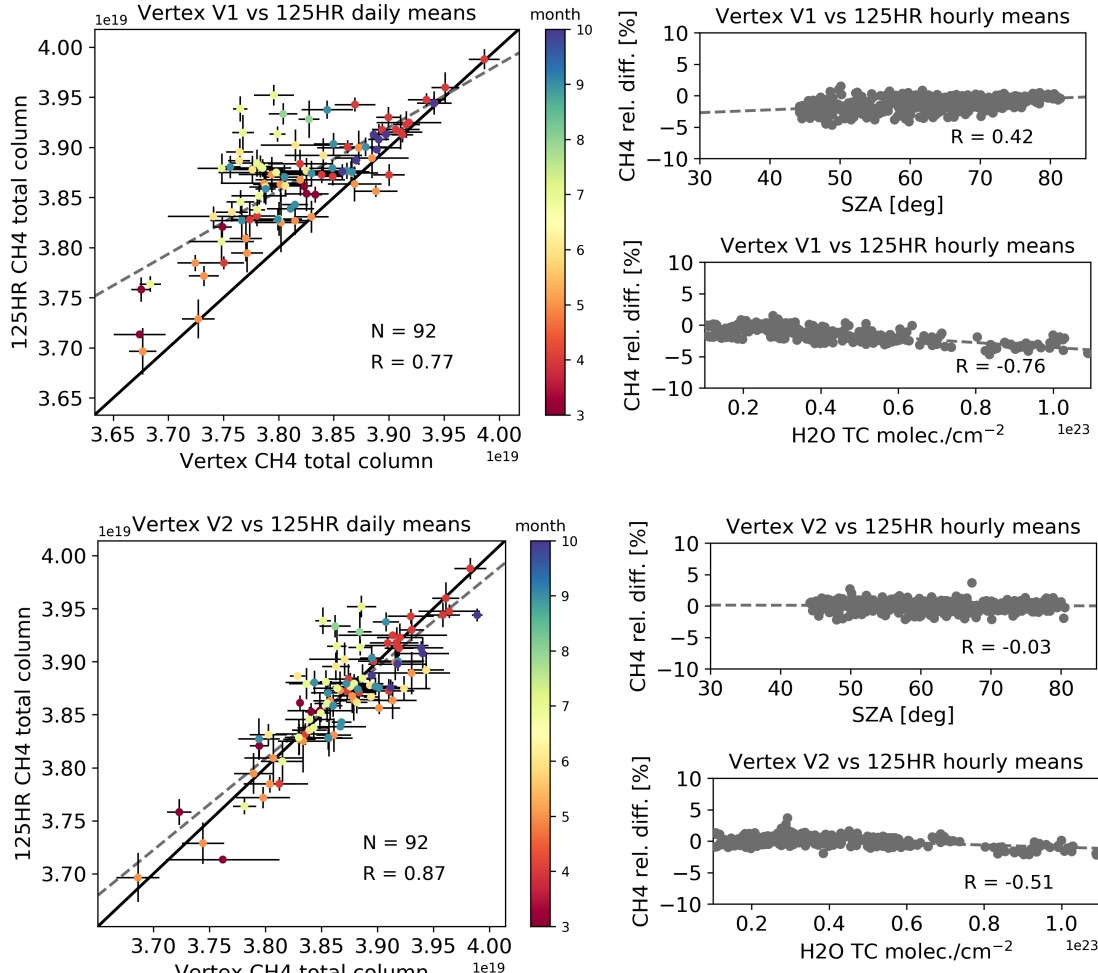

**Figure 10.** Same as Figure 7, but for the co-located 125HR and Vertex 70 V1 (upper) or V2 (lower) CH$_4$ retrievals.

reduced as well (R=-0.51). In addition, the linear regression between the relative differences and H$_2$O total column show that the slope of the regression line for V2 is -1.6 $\times$ 10$^{-23}$, which is about 2.5 times less that for V1 of -3.9 $\times$ 10$^{-23}$.

### 4.2.2 Comparison with AirCore measurements

AirCore consists of a long tube system ascending with a balloon to reach the mid-stratosphere ($\sim$ 25-30 km). It starts collecting the surrounding air when descending from the high altitude to the surface. After landing, the sampled air inside the long tube is passed through by a gas analyzer and represents a vertical profile of the atmospheric components if the analysis is done before the air gets mixed by diffusion. For more information about the AirCore system, we refer to Karion et al. (2010). AirCore launches have been regularly operated by FMI at Sodankylä since 2013, and the data has been used to compare with co-located



FTIR and satellite measurements (Zhou et al., 2019a; Tu et al., 2020; Sha et al., 2020). There are 6 AirCore measurements in 2019 performed in the framework of the FRM4GHG campaign. For the moment, only $CO_2$, $CH_4$ and CO are observed by the AirCore system at Sodankylä.

On the AirCore launch days, the daily mean FTIR retrieved $CH_4$ columns and the relative differences between the FTIR
retrieved $CH_4$ total column and AirCore measurements are listed in Table 4. There is no Vertex measurement on 28 June 2019, and no 125HR measurement on 1 August 2019. To account for the limited vertical sensitivity of the FTIR retrievals, we smooth the AirCore profile with the corresponding FTIR averaging kernels (Rodgers and Connor, 2003).

$$x_{core,s} = x_a + \mathbf{A}(x_{core} - x_a), \tag{5}$$

where $x_{core,s}$ is the smoothed AirCore profile, $x_{core}$ is the extended AirCore profile, $x_a$ is the a priori profile of the FTIR
retrieval, and $\mathbf{A}$ is the FTIR averaging kernel matrix (125HR, Vertex V1 or Vertex V2). Note that the original AirCore profile is extended with a scaled FTIR a priori profile above its maximum height to get the $x_{core}$. The scaling factor is calculated by the ratio of the observed AirCore $CH_4$ and FTIR a priori $CH_4$ mean mole fractions between 20 km and the AirCore profile's maximum height.

The mean relative differences between the FTIR retrieved $CH_4$ total columns and the smoothed AirCore measurements are
0.46%, -1.90% and 0.13% for 125HR, Vertex V1 and Vertex V2, respectively. Consistent with the results shown in Figure 9, the Vertex V1 retrieved $CH_4$ total columns are generally close to the 125HR retrievals in spring, but they become much lower than the 125HR retrievals in summer. As an example, the Vertex V1 retrieved $CH_4$ daily mean is 3.26% less than the smoothed AirCore on 24 July 2019. Several tests have been carried out to understand why there is an underestimation in Vertex V1 retrievals on that day: i) using ERA5 reanalysis data instead of NCEP for the water vapor and temperature a prior profiles;
ii) using the co-located AirCore measurement instead of the WACCM model as the a priori $CH_4$ profile; iii) using the Vertex V2 retrieved $CO_2$, $H_2O$ and HDO instead of WACCM model as the a priori interfering species; iv) using the HITRAN2001 or HITRAN2016 instead of the ATM19 for the spectroscopy. However, all these parameters only have a small impact on the Vertex V1 retrieved $CH_4$ total column, with a relative difference of less than 0.2%. So far, it remains unclear to us why the Vertex V1 retrievals underestimate the total columns as observed by the co-located AirCorements, but a similar bias is observed
with respect to the 125HR retrievals. For the moment, we recommend to use the V2 $CH_4$ retrieval strategy for analysing the Vertex 70 MIR low spectral resolution spectra.

Figure 11 shows the $CH_4$ total columns retrieved from 125HR and Vertex V2, together with the AirCore measurements on these six days. We fit the co-located AirCore and FTIR data pairs using $y = ax$. The slope is 0.9952±0.0041 for 125HR, and 0.9990±0.0048 for Vertex V2. In terms of the vertical profiles, the mean relative difference between the smoothed AirCore and
125HR retrieved $CH_4$ is within 2% for all heights. However, the mean relative difference between the smoothed AirCore and Vertex V2 retrieved $CH_4$ is about -3.0% in the lower troposphere and larger than 5.0% in the stratosphere. The low DOF value (1.3) suggests that we could only get the total column of $CH_4$ from the Vertex V2 retrieval, and the AirCore measurements confirm that the $CH_4$ vertical profile of the 125HR retrieval is more close to the observed one as compared to the Vertex V2 retrieval.





**Table 4.** The daily mean FTIR retrieved CH$_4$ total column ($10^{19}$ molec./cm$^{-2}$), the smoothed AirCore and their relative differences (FTIR-AirCore)/AirCore $\times$ 100%.

| Date | 125HR | Difference | Vertex V1 | Difference | Vertex V2 | Difference |
|---|---|---|---|---|---|---|
| 10-04-2019 | 3.922 | 0.54% | 3.916 | -0.01% | 3.920 | 0.12% |
| 28-06-2019 | 3.803 | 0.17% | | | | |
| 24-07-2019 | 3.884 | 0.01% | 3.780 | -3.26% | 3.886 | 0.21% |
| 01-08-2019 | | | 3.817 | -0.99% | 3.877 | 0.82% |
| 28-08-2019 | 3.933 | 1.15% | 3.804 | -2.85% | 3.862 | -0.65% |
| 09-09-2019 | 3.937 | 0.44% | 3.844 | -2.67% | 3.907 | -0.16% |
| Mean±std | 3.90±0.05 | 0.46±0.39% | 3.83±0.05 | -1.90±1.2% | 3.89±0.02 | 0.13±0.50% |

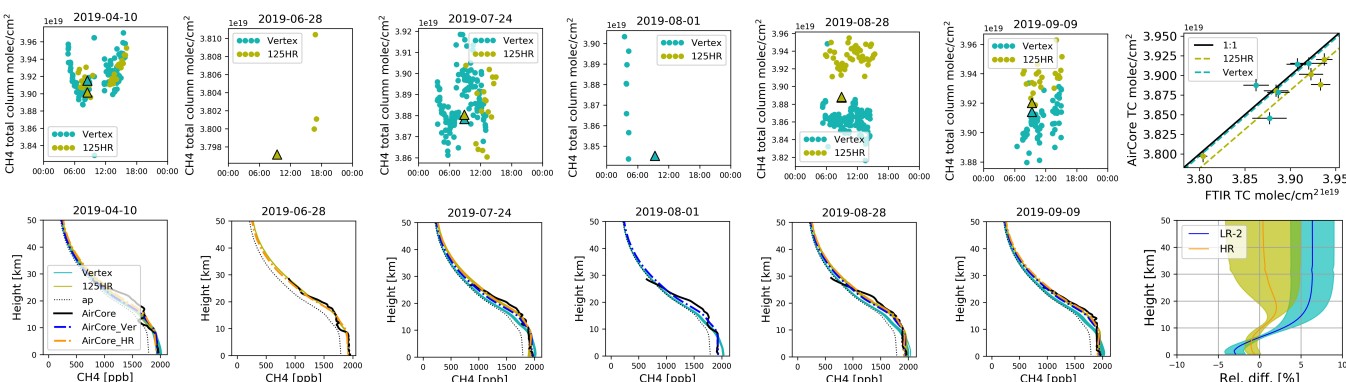

**Figure 11.** Upper panels: the time series of the CH$_4$ total columns retrieved from 125HR (yellow) and Vertex 70 V2 (cyan), together with their corresponding smoothed AirCore total columns (triangle) on the six days. The rightmost plot is the scatter plot between the smoothed AirCore and FTIR CH$_4$ total columns. Note that there is no Vertex measurement on 28 June 2019, and no 125HR measurement on 1 August 2019. On 28 August 2019, the AirCore smoothed (with 125HR and Vertex V2) CH$_4$ total columns are very close to each other so that we can only see one triangle. Lower panels: the CH$_4$ vertical profiles from 125HR and Vertex 70 V2 retrievals, together with the AirCore measurements and the smoothed AirCore profiles. The rightmost plot shows the mean and std of the relative differences between the smoothed AirCore and FTIR CH$_4$ vertical profiles.

### 4.3 N$_2$O vs. CH$_4$

The vertical profile shapes of N$_2$O and CH$_4$ are similar: with a high but relatively constant mole fraction in the troposphere, and decreasing with altitude in the stratosphere. Most of the N$_2$O and CH$_4$ emissions are near the surface so that the mole fractions of these two species are high in the troposphere. The N$_2$O and CH$_4$ molecules are transported from the troposphere to the stratosphere via the Brewer-Dobson circulation. The long lifetimes of N$_2$O and CH$_4$ cause a compact correlation between their





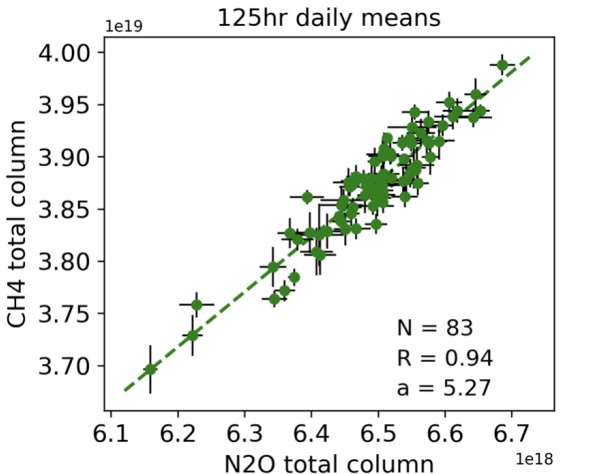
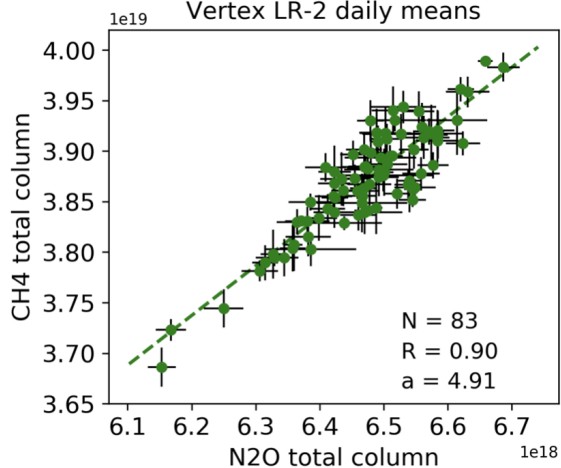

**Figure 12.** The correlation plot between the $N_2O$ and $CH_4$ total column daily means from 125HR (left) and Vertex V2 (right). The error bar is the daily std. N is the number of days, R is the correlation coefficient, and a is the slope of the linear fit.

abundances in the stratosphere, as has been confirmed by balloon, aircraft and satellite measurements (Herman et al., 1998; Sawa et al., 2015; Wang et al., 2014). The 125HR and Vertex 70 measurements show that the $N_2O$ and $CH_4$ total columns have a positive correlation (Figure 12), with an R larger than 0.9. To reduce the sampling uncertainty, we select the days for which both 125HR and Vertex retrievals are available (83 in total). A larger scatter exists in the $N_2O$ and $CH_4$ total columns from

Vertex 70 as compared to 125HR, because of the larger random uncertainties of Vertex 70 retrievals (Tables 2 and 3). The slope of the linear regression ($y = ax + b$) is 5.27±0.20 from the 125HR retrievals and 4.91±0.29 from the Vertex 70 retrievals. The uncertainty of the slope is calculated by the bootstrapping method (Fox, 2008) using the daily std as the random variability. The difference in the slopes between the $N_2O$ and $CH_4$ total columns observed by the 125HR and Vertex 70 measurements are within their combined uncertainties. To conclude, the high correlations between $N_2O$ and $CH_4$ total columns at Sodankylä are

confirmed by the 125HR and Vertex 70 retrievals.

## 5   Conclusions

In this study, the atmospheric abundances of $N_2O$ and $CH_4$ are retrieved from low spectra-resolution (0.2 cm$^{-1}$) MIR solar absorption spectra recorded by the Bruker Vertex 70 spectrometer in 2019 at Sodankylä, and compared with standard NDACC retrievals from co-located 125HR spectra with a spectral resolution of 0.005 cm$^{-1}$. Moreover, the $CH_4$ retrievals from the

125HR and Vertex 70 spectra are also compared with the co-located in situ AirCore measurements.

We find that the standard NDACC retrieval strategy works well for the Vertex 70 $N_2O$ retrieval as well. The relative differences between the $N_2O$ total columns retrieved from the 125HR and Vertex 70 spectra are -0.3±0.7%. The correlation between the $N_2O$ total columns retrieved from 125HR and Vertex 70 spectra is 0.93, and the relative differences are almost independent





of SZA or water vapour content. However, the retrieval uncertainties of the N$_2$O total columns from the Vertex 70 spectra are larger than those of the total columns retrieved from 125HR spectra due especially to a higher measurement noise. What's more, the mean DOF of the N$_2$O retrievals from the Vertex 70 spectra is 1.6, which is less than the DOF of 2.6 obtained for the retrievals from the 125HR spectra. This agrees with the expectation that less vertical information can be derived from the

spectra with a lower spectral resolution. Inspection of the N2O partial columns (0-6 km and 6-25 km) confirms that this vertical information is captured by the Vertex 70 measurements, albeit with a larger uncertainty than what is obtained from the 125HR measurements.

   Unlike N$_2$O, the recommended NDACC retrieval strategy does not perform well for the CH$_4$ retrievals from Vertex 70 spectra. Using the same retrieval windows (V1 retrieval strategy), the relative differences between the CH$_4$ total columns

retrieved from the 125HR and Vertex 70 spectra are -1.3±1.1%, and show a seasonal variation: in spring the Vertex V1 retrieved CH$_4$ total columns are close to the ones retrieved from the 125HR spectra but in summer they are about 3.0% smaller than the 125HR retrievals. In addition, these relative differences vary with SZA and water vapour content. The DOF of V1 retrieved CH$_4$ profiles from Vertex 70 spectra is 1.0. The correlation coefficient between the CH$_4$ total columns retrieved from 125HR and Vertex 70 spectra is 0.77. The underestimation of the Vertex V1 CH$_4$ retrieved columns in summer is confirmed by the

co-located AirCore profiles. In this study, we propose to replace two microwindows of the recommended NDACC retrievals strategy (2835.55-2835.8 cm$^{-1}$ and 2903.82-2903.925 cm$^{-1}$) by two alternative ones, namely 2875.15-2875.85 cm$^{-1}$ and 2898.3-2898.93 cm$^{-1}$, including stronger CH$_4$ absorption lines, to perform the CH$_4$ profile retrievals from the Vertex 70 spectra (V2 retrieval strategy). The DOF of Vertex V2 CH$_4$ retrievals is 1.3, providing more vertical information as compared to Vertex V1 retrievals. The relative differences between the CH$_4$ total columns retrieved from 125HR and Vertex 70 spectra are

0.0±0.8%, with a much reduced seasonal variation. The correlation coefficient between the CH$_4$ total columns retrieved from 125HR and Vertex 70 spectra increases to 0.87. The relative differences between the CH$_4$ total columns from the co-located AirCore and Vertex V2 retrievals are 0.13±0.50%.

   The NIR vs. MIR and low vs. high resolution performance differences are an interesting measurement techniques topic to characterize, including how performance differences have evolved over time, with various technological and retrieval advance-

ments. In the framework of the ESA FRM4GHG-2 project, we intend to design a compact solar tracker system and enclosure to make the Bruker Vertex 70 even more convenient to travel and operate autonomously. More campaigns will be carried out to better understand the performances of the CH$_4$ and N$_2$O Vertex retrievals with different conditions, such as at a humid or a low-latitude site. Such kind of portable FTIR will help to fill the gaps of the NDACC network for CH$_4$ and N$_2$O observations and contribute effectively to satellite validation, model verification, and other scientific studies. In the future, it is also planned

to investigate the retrieval of additional tropospheric species from the Bruker Vertex 70 low resolution MIR spectra.

*Data availability.* The Bruker 125HR retrievals at Sodankylä are publicly available at NDACC archive (https://www-air.larc.nasa.gov/missions/ndacc/). The Vertex 70 retrievals and AirCore profiles are available upon request to the authors.



*Competing interests.* The authors declare that they have no conflict of interest.

*Acknowledgements.* This study was supported by European Space Agency projects FRM4GHG and FRM4GHG2.0, which received research funding from ESA's FRM programme under grant agreement no. 4000117640/16/I-LG and 4000136108/21/I-DT-lr.

*Author contributions.* MZ wrote the manuscript. MZ, MKS, BL, JN and MDM designed the experiment. CH, RK, PH, CP contributed to the

5   FTIR measurements at Sodankylä. RK and HC provided the AirCore measurements. All the authors read and commented on the manuscript.



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
