# Peer review of "Atmospheric N2O and CH4 total columns retrieved from low-resolution FTIR spectra (Bruker Vertex 70) in the mid-infrared region"

_Atmospheric Measurement Techniques, 2022_

## Referee Comment (RC1)

**General Comments:**

The paper presents a comparison of N2O and CH4 total and partial columns retrieved from low and high resolution FTIR spectra over the course of 8 months. The work is well motivated, technically sound, well written and easy to follow.    It is interesting and within the scope of AMT.

**Major uncertainty:**

"Note that the observed spectrum with a high spectral resolution (125HR) is not well fitted in band 3V2. **Therefore, we do not apply the V2 window choice for 125HR spectra.**"    While it's clear from F3 that CH4 in band 3V2 is not fitted well in 125HR spectra, the fit is not much worse than in band 5 (also due to CH4 problems), which is part of the regular NDACC retrieval -- but maybe shouldn't be?    What's surprising and problematic is that the regular NDACC retrieval column kernel (F5) is not close to 1.0 throughout the troposphere, as is the case for N2O (F4).    I see the authors tried many microwindows in their work to replace bands 3 and 4, which contain only 4 and 2 spectral samples, but were the column kernels of the 125HR a driving factor in the analysis?      We need (??) band 1 and 6 for HDO and H2O, respectively, but how would band 2 (70% CH4 transmittance) + band 5 (55% CH4 transmittance) + band 4V2 (30% CH4 transmittance) look in terms of the 125HR column kernel?    The range of CH4 absorption depths should provide good sensitivity to both lower and upper atmospheric partial columns.    My concern is that by comparing 125HR and V70-V2 retrievals performed in different microwindows, **it may be that the agreement is improved from V70-V1 for the wrong reasons, i.e., because V1 is doomed to fail on account of band 3 and 4 while V2 is never applied to 125HR spectra.**    Also/finally, it would be good to have a regular NDACC retrieval combination that made the column kernel close to 1 in the troposphere - can bands 1+6 (HDO, H2O) combined with 2+5+4V2 achieve this?    Maybe band 5 can be dropped even, since it's not fit so well either in 125HR spectra?    Have the **weighting functions** been examined to show the vertical response in each band?    Could the relative difference w.r.t. smoothed **AirCore** measurements be improved from 0.46% for 125HR to something lower like 0.13% for V70-V2??

**Minor comments:**

In a preceding study (Sha et al., 2020) the V70 was studied along with the EM27/Sun and IRCube to determine performance in NIR retrievals of CH4 CO and CO2.    Why was Bruker's Vertex 90 chosen for the analogous MIR work out of these 3 low resolution options?    Please add these considerations to the discussion on P2 around L29.
Is lowest cost the determinant, or is it in fact the Vertex 90's highest spectral resolution (0.2 cm-1), which provides the greatest partial column information over the other two low-res options (0.5 cm-1)?

Some deeper discussion of the microwindows used by Taylor et al. (2008) is warranted, give that it is the only other MIR low vs. high resolution study cited.    P2L32 cites "poor correlations" for CH4 and N2O in that study but actually the cited N2O result is very similar at -0.36% c.f. -0.30% in this study.    Their CH4

result was 3.7% compared to -1.3% (V1) and 0.0% (V2) in this study -- are the microwindows different?

P13    F7 (left) and also P16 F10 (left):    why the switch from hourly mean relative differences to daily mean total column correlations?    What is N and R for hourly mean total column correlation plots (without error bars necessary)?

P14L10    "they compensate" --> some of this compensation is due to DOFS = 0.8 (neither layer is truly indepdendent) so I wonder about accuracy of statement that "partial columns [...] can be retrieved with confidence [...] albeit with a larger uncertainty."

P19L2    "total columns" --> since we know that stratospheric CH4 and N2O are well correlated, why not loook at partial columns in F12 instead of total columns?    I realize now that CH4 DOFs do not permit it for V70-v2, but would the correlation be even more compact for 125HR partial columns?

**Technical corrections:**

Sometimes "Vertex 70" is used and a few times just "Vertex" is used while "125HR" is used throughout for the high res measurements and retrievals.    It would be easier to follow and more consistent if the compact term "V70" was introduced and followed throughout the manuscript, later expanding to V70-v1 and V70-v2, as it comes up.

P1L9-L14    include 1sigma also with CH4 results, not just N2O
P1L12    remove "an"
P1L17    remove "an"
P1L18    change "instruments" to "instrument"

P2L14-L16    remove four instances of "the"

P3L7    Reference to Pougatchev et al., 1995 not appropriate for SFIT4

P6L6    "4 and 2 wavenumbers" --> "4 and 2 samples in wavenumber space"

P7L17    "takes up only" --> "makes up only"
P7    What % of CH4 total column above 10 km?    Is it also around 1.5% like for N2O?

P8    F3 caption "mirco" --> "micro" (twice)

P9    F4 caption "mvr" --> "vrm"
P9L13 and onward:    while "std" is defined on P7, the greek symbol \sigma seems more appropriate
P9L7 to P11L2    provide some references supporting the random and systematic error choices in the budget analysis

P10    F5 caption "mvr" --> "vmr"

P11L3    "profiles" --> "columns"
P11    T2 caption "tet" --> "text"
P11L9-10    the random uncertainties show significant contribution from SZA, especially for 125HR N2O, but SZA is not discussed at all in this discussion of T2 and T3

P12L5    consider changing "co-located" to "coincident" here and throughout, since the measurements are always co-located, but not always coincident

P13    F6 caption "from their" --> "between their"

P14L1    "correction" --> "correlation"
P14L2    "respectively."    --> "respectively (not shown)." because there's no plot like F7 (left)
P14L8    "What's more" --> "Moreover,"
P14 F8 caption "uppder" --> "upper"

P15L3-L4    remove one too many "only"

P16L2    what is the slope change for SZA?
P16L6    remove "by" from "by a gas analyzer"
P16L7    ", we refer to Karion et al. (2010)."    --> "(Karion et al., 2010)." suffices

P17L24    "AirCorements"    --> "AirCore measurements"

P18 T4    "the smoothed AirCore" is never actually given in T4, only the relative difference
P19L12    "low spectra-resolution" --> "low resolution spectra" (no hyphens needed)

---

## Author Comment (AC1)

*Black: referee's comments* *green: authors' answers*
*First of all, we would like to thank the referee for the detailed analysis of our paper.*
*For the details, please look into the paper with keeping track of changes.*

Referee #1

General Comments: The paper presents a comparison of N2O and CH4 total and partial columns retrieved from low and high resolution FTIR spectra over the course of 8 months. The work is well motivated, technically sound, well written and easy to follow. It is interesting and within the scope of AMT.

Major uncertainty:
"Note that the observed spectrum with a high spectral resolution (125HR) is not well fitted in band 3V2. Therefore, we do not apply the V2 window choice for 125HR spectra." While it's clear from F3 that CH4 in band 3V2 is not fitted well in 125HR spectra, the fit is not much worse than in band 5 (also due to CH4 problems), which is part of the regular NDACC retrieval -- but maybe shouldn't be? What's surprising and problematic is that the regular NDACC retrieval column kernel (F5) is not close to 1.0 throughout the troposphere, as is the case for N2O (F4). I see the authors tried many microwindows in their work to replace bands 3 and 4, which contain only 4 and 2 spectral samples, but were the column kernels of the 125HR a driving factor in the analysis? We need (??) band 1 and 6 for HDO and H2O, respectively, but how would band 2 (70% CH4 transmittance) + band 5 (55% CH4 transmittance) + band 4V2 (30% CH4 transmittance) look in terms of the 125HR column kernel? The range of CH4 absorption depths should provide good sensitivity to both lower and upper atmospheric partial columns. My concern is that by comparing 125HR and V70-V2 retrievals performed in different microwindows, it may be that the agreement is improved from V70-V1 for the wrong reasons, i.e., because V1 is doomed to fail on account of band 3 and 4 while V2 is never applied to 125HR spectra. Also/finally, it would be good to have a regular NDACC retrieval combination that made the column kernel close to 1 in the troposphere - can bands 1+6 (HDO, H2O) combined with 2+5+4V2 achieve this? Maybe band 5 can be dropped even, since it's not fit so well either in 125HR spectra? Have the weighting functions been examined to show the vertical response in each band? Could the relative difference w.r.t. smoothed AirCore measurements be improved from 0.46% for 125HR to something lower like 0.13% for V70-V2??

Thanks for the suggestions and comments.

The choice of CH4 retrieval is always interesting and challenging because there are many CH4 absorption lines in the infrared region, as well as many interfering species. The NDACC retrieval windows are recommended by the IRWG community and have been used widely used within the network (Sepúlveda et al., 2014; Zhou et al., 2018). The column averaging kernel using the standard NDACC CH4 windows at Sodankyla is 0.7 - 0.8 near the surface (Figure 5), which is consistent with other sites, e.g. Reunion Island, Izana, and Jungfraujoch. The reason is that the CH4 absorption lines used in the standard NDACC retrieval windows are relatively weak, as you can see that the lowest transmittance of the CH4 retrieval window is about 0.5 (Figure 3). The column-averaging kernel of N2O is pretty close to 1 at all levels because the N2O absorption lines used in the standard NDACC retrieval windows are much stronger than CH4. To conclude, when the absorption line of the target species is weak, we have less

sensitivity in the troposphere. With band2 + band 5 + band 4V2, the column averaging kennel from the 125HR high-resolution spectrum shows a slightly better sensitivity in the troposphere, but the retrieved CH4 total column will change too (~2%). The retrieval uncertainty is not only determined by the column averaging kernel but also affected by the fitting residual, spectroscopic uncertainty, and many other retrieval parameters. If we change the retrieval window, the retrieved NDACC CH4 profiles/columns will change too. We believe that it might be some room left for the whole IRWG group to improve the standard NDACC CH4 retrieval window to have better sensitivity in the troposphere, and a better residual in the fitting, but this is beyond the scope of this study. As the NDACC CH4 retrievals at Sodankyla are publicly available in the NASA archive and they are currently applied for the TROPOMI validation and the ECMWF CAMS model validation, we do not want to change the standard NDACC CH4 retrieval settings.

"My concern is that by comparing 125HR and V70-V2 retrievals performed in different microwindows, it may be that the agreement is improved from V70-V1 for the wrong reasons, i.e., because V1 is doomed to fail on account of band 3 and 4 while V2 is never applied to 125HR spectra."

Using different retrieval windows for 125HR and Vertex respectively is acceptable because of their different spectral resolutions. The comparison with AirCore profiles (after smoothing correction) confirms that the Vertex v70-V2 shows a better agreement. Moreover, the systematic and random smoothing errors are estimated to be 0.1% and 0.3% for NDACC CH4 total column, 0.2% and 0.5% for the Vertex V1 CH4 total column, and 0.1% and 0.1% for the Vertex V2 CH4 total column, respectively. Based on the smoothing estimations, we understand that the smoothing error is much smaller than the differences, and the change of the averaging kernel from v70-V1 to v70-V2 cannot make such an improvement in their difference of -1.3±1.1% to 0.0±0.8% against with NDACC measurements.

Minor comments:

In a preceding study (Sha et al., 2020) the V70 was studied along with the EM27/Sun and IRCube to determine performance in NIR retrievals of CH4 CO and CO2. Why was Bruker's Vertex 90 chosen for the analogous MIR work out of these 3 low resolution options? Please add these considerations to the discussion on P2 around L29. Is lowest cost the determinant, or is it in fact the Vertex 90's highest spectral resolution (0.2 cm-1), which provides the greatest partial column information over the other two low-res options (0.5 cm-1)?

Done.

The MIR measurements were operated by the Vertex 70v but not by the EM27/sun and the IRCube. The reason is the lowest cost/manpower reason. We need much more effects to make the MIR measurement work for the EM27/sun and the IRCube.

Some deeper discussion of the microwindows used by Taylor et al. (2008) is warranted, give that it is the only other MIR low vs. high resolution study cited. P2L32 cites "poor correlations" for CH4 and N2O in that study but actually the cited N2O result is very similar at -0.36% c.f. -0.30% in this study. Their CH4 result was 3.7% compared to -1.3% (V1) and 0.0% (V2) in this study -- are the microwindows different?

Thanks, more discussions are added now.

P13 F7 (left) and also P16 F10 (left): why the switch from hourly mean relative differences to daily mean total column correlations? What is N and R for hourly mean total column correlation plots (without error bars necessary)?
The correlations using hourly means or daily means are similar (Figure A1). The hourly means are plotted here to have a better view of the measurement month.

[Figure]

Figure A1. The correlation plots of the 125HR and Vertex 70 retrieved N2O (top) and CH4 (bottom) columns for their coincident daily means (left) and hourly means (right). The Vertex 70 retrievals use the V2 retrieval micro-windows.

P14L10 "they compensate" --> some of this compensation is due to DOFS = 0.8 (neither layer is truly indepdendent) so I wonder about accuracy of statement that "partial columns [...] can be retrieved with confidence [...] albeit with a larger uncertainty."
Thanks for the comment. Agree with the referee that DOFS = 0.8 indicates that neither layer is truly independent. We change the sentence to "Keep in mind that the DOFs of the two partial columns (0-6 km and 6-25 km) of $N_2O$ derived from the Vertex 70 spectra are both 0.8, indicating that neither layer is truly independent. In summary, the partial columns (0-6 km and 6-25 km) of $N_2O$ can be derived from the Vertex 70 spectra, but with a larger uncertainty as compared to the 125HR retrievals.

P19L2 "total columns" --> since we know that stratospheric CH4 and N2O are well correlated, why not look at partial columns in F12 instead of total columns? I realize now that CH4 DOFs do not permit it for V70-v2, but would the correlation be even more compact for 125HR partial columns?

Thanks for the comments. Indeed, the DOFs of V70-v2 (~1.3) only allows us to use the total column. We plot the partial columns of CH4 and N2O from the 125HR here (6-30 km). The correlation of the partial column is better, but it is not comparable with the Vertex retrievals (so we not show them in the manuscript).

[Figure]

Figure A2. The correlation plots between the 125HR N2O and CH4 retrievals for total column (left), for partial column in 0-6 km (middle), and for partial column in 6-30 km (right).

Technical corrections:

Sometimes "Vertex 70" is used and a few times just "Vertex" is used while "125HR" is used throughout for the high res measurements and retrievals. It would be easier to follow and more consistent if the compact term "V70" was introduced and followed throughout the manuscript, later expanding to V70-v1 and V70-v2, as it comes up.
Corrected, we use Vertex 70 consistently now.
P1L9-L14 include 1sigma also with CH4 results, not just N2O
Done.
P1L12 remove "an"
Done.
P1L17 remove "an"
Done.
P1L18 change "instruments" to "instrument"
Done.
P2L14-L16 remove four instances of "the"
Done.
P3L7 Reference to Pougatchev et al., 1995 not appropriate for SFIT4
We added the Hase 2004 here too. The paper of the SFIT4 retrieval code is in preparation.
P6L6 "4 and 2 wavenumbers" --> "4 and 2 samples in wavenumber space"
Done.
P7L17 "takes up only" --> "makes up only"
Done.
P7 What % of CH4 total column above 20 km? Is it also around 1.5% like for N2O?
According to the a priori profile, the CH4 partial column above 20 km accounts for 2.5%.
P8 F3 caption "mirco" --> "micro" (twice)
Corrected.
P9 F4 caption "mvr" --> "vrm"
Corrected.
P9L13 and onward: while "std" is defined on P7, the greek symbol \sigma seems more appropriate
We prefer to keep the std in the text.

P9L7 to P11L2 provide some references supporting the random and systematic error choices in the budget analysis

Added.

P10 F5 caption "mvr" --> "vmr"

Corrected.

P11L3 "profiles" --> "columns"

Done.

P11 T2 caption "tet" --> "text"

Corrected.

P11L9-10 the random uncertainties show significant contribution from SZA, especially for 125HR N2O, but SZA is not discussed at all in this discussion of T2 and T3

Added.

P12L5 consider changing "co-located" to "coincident" here and throughout, since the measurements are always co-located, but not always coincident

Done.

P13 F6 caption "from their" --> "between their"

Done.

P14L1 "correction" --> "correlation"

Done.

P14L2 "respectively." --> "respectively (not shown)." because there's no plot like F7 (left)

Done.

P14L8 "What's more" --> "Moreover,"

Done.

P14 F8 caption "uppder" --> "upper"

Done.

P15L3-L4 remove one too many "only"

Done.

P16L2 what is the slope change for SZA?

Added more info.

P16L6 remove "by" from "by a gas analyzer"

Done.

P16L7 ", we refer to Karion et al. (2010)." --> "(Karion et al., 2010)."

Done.

P17L24 "AirCorements" --> "AirCore measurements"

Done.

P18 T4 "the smoothed AirCore" is never actually given in T4, only the relative difference

Corrected.

P19L12 "low spectra-resolution" --> "low resolution spectra" (no hyphens needed)

Done.

References:

Sepúlveda, E., Schneider, M., Hase, F., Barthlott, S., Dubravica, D., García, O. E., Gomez-Pelaez, A., González, Y., Guerra, J. C., Gisi, M., Kohlhepp, R., Dohe, S., Blumenstock, T., Strong, K., Weaver, D., Palm, M., Sadeghi, A., Deutscher, N. M., Warneke, T., Notholt, J., Jones, N., Griffith, D. W. T., Smale, D., Brailsford, G. W., Robinson, J., Meinhardt, F., Steinbacher, M., Aalto, T., and Worthy, D.: Tropospheric CH4 signals as observed by NDACC FTIR at globally distributed sites and comparison to GAW surface in situ measurements, Atmos. Meas. Tech., 7, 2337–2360, https://doi.org/10.5194/amt-7-2337-2014, 2014.

Zhou, M., Langerock, B., Vigouroux, C., Sha, M. K., Ramonet, M., Delmotte, M., Mahieu, E., Bader, W., Hermans, C., Kumps, N., Metzger, J.-M., Duflot, V., Wang, Z., Palm, M., and De Mazière, M.: Atmospheric CO and CH4 time series and seasonal variations on Reunion Island from ground-based in situ and FTIR (NDACC and TCCON) measurements, Atmos. Chem. Phys., 18, 13881–13901, https://doi.org/10.5194/acp-18-13881-2018, 2018.

---

## Author Comment (AC2)

*Black: referee's comments green: authors' answers*
*First of all, we would like to thank the referee for the detailed analysis of our paper.*
*For the details, please look into the paper with keeping track of changes.*

Referee #2

Dear authors,
first I have to apologize for the long time that this manuscript has spent in review. It was extremely difficult to find reviewers that were not connected with the FRM4GHG project. This project includes a large part of the FTIR community. From the ones I identified, 6 were unavailable, declined or had a conflict of interests. Therefore, I would like to extend special thanks to anonymous reviewer #1 who provided a thorough report on time.
Unfortunately, the whole process took place while I was absorbed by tragic family matters for several months. The backlog that piled up while I was hardly able to work also delayed this review process. I appreciate your understanding that the first author Minqiang Zhou has expressed to me personally already.
Not being able to come up with 2 reviewers in the end, I am forced to resort to an editor's review to finish the process. My own FTIR experience is limited to TCCON hi-resolution retrievals with GGG on the IFS125HR. My experience with NDACC, SFIT4 and low-resolution retrievals is very limited. Therefore, I would only add the minor comments below and leave the detailed corrections to the report already provided by reviewer #1.
Kind regards
Dietrich Feist

First of all, we would like to thank Dietrich Feist and Thomas Wagner for your great help of handling our manuscript. We understand that many relative colleagues are involved in the FRM4GHG-1/2 projects and it is difficult to find the experts. We also heard that it was a very hard time for Dietrich during the last couple months, and we sincerely hope everything gets better now!

Comments:
- I find the naming of the bands 1-6, V1, V2, 3V2, 4V2 confusing. For clarity, please provide a table that describes the major parameters (wave number range, instrument, species retrieved).

Thanks for your comment. We add a table to show the information of these abbreviaiton in the revised version.

- p. 14, l. 7: Do you mean standard deviation with "stds"? Please spell out to avoid confusion.

Std stands for 'standard deviation' and stds stands for 'standard deviations'. It has been clarified now in the revised version.